# Genetic Deletion of AT_1a_ Receptor or Na^+^/H^+^ Exchanger 3 Selectively in the Proximal Tubules of the Kidney Attenuates Two-Kidney, One-Clip Goldblatt Hypertension in Mice

**DOI:** 10.3390/ijms232415798

**Published:** 2022-12-13

**Authors:** Xiao Chun Li, Rumana Hassan, Ana Paula O. Leite, Akemi Katsurada, Courtney Dugas, Ryosuke Sato, Jia Long Zhuo

**Affiliations:** 1Tulane Hypertension and Renal Center of Excellence, 1430 Tulane Avenue, New Orleans, LA 70112, USA; 2Department of Physiology, Tulane University School of Medicine, 1430 Tulane Avenue, New Orleans, LA 70112, USA

**Keywords:** 2-kidney, 1-clip Goldblatt hypertension, angiotensin II, AT_1a_ receptor, NHE3, proximal tubule

## Abstract

The roles of angiotensin II (Ang II) AT_1_ (AT_1a_) receptors and its downstream target Na^+^/H^+^ exchanger 3 (NHE3) in the proximal tubules in the development of two-kidney, 1-clip (2K1C) Goldblatt hypertension have not been investigated previously. The present study tested the hypothesis that deletion of the AT_1a_ receptor or NHE3 selectively in the proximal tubules of the kidney attenuates the development of 2K1C hypertension using novel mouse models with proximal tubule-specific deletion of AT_1a_ receptors or NHE3. 2K1C Goldblatt hypertension was induced by placing a silver clip (0.12 mm) on the left renal artery for 4 weeks in adult male wild-type (WT), global *Agtr1a*^−/−^, proximal tubule (PT)-specific PT-*Agtr1a*^−/−^ or PT-*Nhe3*^−/−^ mice, respectively. As expected, telemetry blood pressure increased in a time-dependent manner in WT mice, reaching a maximal response by Week 3 (*p* < 0.01). 2K1C hypertension in WT mice was associated with increases in renin expression in the clipped kidney and decreases in the nonclipped kidney (*p* < 0.05). Plasma and kidney Ang II were significantly increased in WT mice with 2K1C hypertension (*p* < 0.05). Tubulointerstitial fibrotic responses were significantly increased in the clipped kidney (*p* < 0.01). Whole-body deletion of AT_1a_ receptors completely blocked the development of 2K1C hypertension in *Agtr1a*^−/−^ mice (*p* < 0.01 vs. WT). Likewise, proximal tubule-specific deletion of *Agtr1a* in PT-*Agtr1a*^−/−^ mice or NHE3 in PT-*Nhe3*^−/−^ mice also blocked the development of 2K1C hypertension (*p* < 0.01 vs. WT). Taken together, the present study provides new evidence for a critical role of proximal tubule Ang II/AT_1_ (AT_1a_)/NHE3 axis in the development of 2K1C Goldblatt hypertension.

## 1. Introduction

Hypertension is well recognized as a multifactorial disorder involving genetic, lifestyle, neural, and endocrine factors [1,2,3,4,5,6]. Hypertension is classified into two different forms; one is primary or essential, and the other is of secondary nature [1,2,3,4,5,6]. According to American Heart Association (AHA) and National Heart, Lung, and Blood Institute (NHLBI), most hypertensive patients have essential hypertension without readily identifiable causes [1,3,5,7]. By contrast, about 10% of hypertensive patients have secondary hypertension with identifiable causes and close to other 17% may be classified as apparent treatment resistant hypertension [3,5,7]. Renal vascular hypertension is the most common secondary hypertension with identifiable causes [3,5,7,8,9]. In humans, the most significant etiology for renal vascular hypertension is atherosclerosis as the major cause of renal artery stenosis. By contrast, the mechanisms responsible for the development of renal vascular hypertension in animals have been extensively studied using rat and mouse models of two-kidney, one-clip (2K1C) Goldblatt hypertension previously [10,11,12,13,14,15,16,17,18]. 2K1C Goldblatt hypertension in animals is induced primarily by mechanically narrowing one renal artery with a silver clip placed around one of the renal arteries, which induces renal ischemia and decreases renal blood flow and glomerular filtration [10,11,12,13,14,15,16,17].

It is well-recognized that the activation of the renin-angiotensin system (RAS) in the clipped kidney is the most important mechanism contributing to the development of 2K1C Goldblatt hypertension [18,19,20,21,22,23,24,25]. Indeed, most if not all studies have shown that the development of 2K1C Goldblatt hypertension is associated with marked increases in renin expression and angiotensin II (Ang II) generation in the clipped kidney and subsequently their release into the circulation, leading increases in the circulating Ang II level [10,15,18,25]. There is a consensus that the increases in the circulating Ang II by clipping one of the kidneys induces hypertension by a complexity of pathways, including but not limited to causing vascular smooth muscle contraction [26,27,28], elevating central and peripheral sympathetic nerve activities [14,29,30,31], and promoting aldosterone release [29]. The development of 2K1C Goldblatt hypertension is often blocked or attenuated by systemic blockade of AT_1_ receptors by Ang II receptor blockers (ARBs) in animal models, establishing a key role of the AT_1_ receptor in the pathogenesis of 2K1C Goldblatt hypertension [10,11,12,13,14,15,16,22]. Subsequent studies by Cervenka et al. and others have further shown that whole-body deletion of AT_1a_ receptors blocked the development of 2K1C hypertension in *Agtr1a*^−/−^ mice, supporting a key role of systemic AT_1a_ receptors in the development of 2K1C hypertension [15,16,26,27,28]. Nevertheless, the use of whole-body *Agtr1a*^−/−^ mouse model to determine the mechanisms of 2K1C hypertension is similar to the use of ARBs because both approaches block AT_1_ (AT_1a_) receptors in every tissue. Thus, these whole-body approaches are therefore unlikely to identify and differentiate the roles or contributions of AT_1_ (AT_1a_) receptors between the kidney and extrarenal tissues, and those of AT_1_ (AT_1a_) receptors between glomerular, vascular, and tubular epithelial cells in the kidney.

Against this background, the present study was designed to determine the specific roles of the AT_1_ (AT_1a_) receptor and its downstream target Na^+^/H^+^ exchanger 3 (NHE3) in the proximal tubules in the development of 2K1C Goldblatt hypertension using novel mouse models with proximal tubule-specific deletion of AT_1a_ receptors or NHE3, as we reported recently [32,33,34,35]. We hypothesized that in normal wild-type mice, clipping the left renal artery leads to activation of intratubular RAS in the clipped kidney with increased expression of the rate-limiting enzyme renin and generation of Ang II in the proximal tubules, which in turn induces AT_1_ (AT_1a_) receptor-mediated, NHE3-dependent stimulation of proximal tubule Na^+^ reabsorption leading to the development of 2K1C Goldblatt hypertension. We reasoned that if this hypothesis is correct, then genetic deletion of AT_1_ (AT_1a_) receptors or NHE3 selectively in the proximal tubules would attenuate or prevent the development of 2K1C Goldblatt hypertension in proximal tubule-specific PT-*Agtr1a*^−/−^ or PT-*Nhe3*^−/−^ mice.

## 2. Results

### 2.1. Basal Physiological Phenotypes and Their Responses to the Development of 2K1C Goldblatt Hypertension in All Strains of Mice

Table 1 summarizes basal physiological phenotypes of adult male WT, global *Agtr1a*^−/−^, and proximal tubule-specific PT-*Agtr1a^−/−^,* and PT*-Nhe3^−/−^* mice. Table 2 summarizes the physiological phenotypic responses to the development of 2K1C Goldblatt hypertension in adult male WT, global *Agtr1a*^−/−^, proximal tubule-specific PT-*Agtr1a*^−/−^, and PT*-Nhe3^−/−^* mice. Under basal conditions, the heart wt. (*p* < 0.05) or the kidney to body wt. ratios decreased significantly in whole-body *Agtr1a*^−/−^ mice, compared with WT controls (*p* < 0.01). However, there were no differences in the heart wt. or the kidney to body wt. ratios between WT and proximal tubule-specific PT-*Agtr1a*^−/−^, or PT*-Nhe3^−/−^* mice (*n.s*.). Whole-body or proximal tubule-specific deletion of AT_1a_ receptors or NHE3 in the kidney induced significant increases in 24 h urinary sodium (Na^+^) and potassium (K^+^) excretion (Table 1). In response to the induction of 2K1C Goldblatt hypertension, 24 h urinary sodium, potassium, and chloride (Cl^−^) excretion were also significantly increased compared with WT mice (Table 2).

### 2.2. Effects of Induction of 2K1C Goldblatt Hypertension on Whole-Kidney Glomerular Filtration Rate in All Mouse Models

Figure 1 shows that induction of 2K1C Goldblatt hypertension by placing a 0.12 mm silver clip on the left renal artery of the kidney significantly decreased glomerular filtration rate (GFR) by >40% similarly in WT, whole-body *Agtr1a*^−/−^, PT- *Agtr1a*^−/−^, and PT-*Nhe3*^−/−^ mice (*p* < 0.01). These data suggest that all animals developed a similar decrease in renal blood perfusion to the left kidney in all groups of mice.

### 2.3. AT_1_ (AT_1a_) Receptor-Mediated, Ang II-Dependent Development of 2K1C Goldblatt Hypertension in WT Mice

Figure 2A shows the time-dependent development of 2K1C Goldblatt hypertension in adult male WT mice with an 0.12 mm-clip in the left kidney and telemetry systolic blood pressure increasing from baseline of 112 ± 3 mmHg (n = 15) to 125 ± 2 mmHg one week after the induction of 2K1C hypertension (*p* < 0.01). Systolic blood pressure continued to increase in a time-dependent manner reaching a peak response during the 3rd week (149 ± 3 mmHg, *p* < 0.01) and sustained until the end of the experiment at Week 4 (148 ± 5 mmHg, *p* < 0.01) (Figure 2A). The increases in diastolic and mean arterial blood pressure are similar in the magnitudes of systolic blood pressure.

### 2.4. Activation of Circulating and Intratubular RAS in WT Mice with 2K1C Goldblatt Hypertension

The development of 2K1C Goldblatt hypertension is associated with marked increases in renin expression and angiotensin II (Ang II) generation in the clipped kidney and reportedly in the nonclipped kidney [10,15,16,25]. As expected, renin mRNA expression was significantly increased in the clipped kidney by 57% at the end of experiment (Control: 2366 ± 255 copies/ng RNA vs. 2K1C: 3716 ± 549 copies/ng RNA, *p* < 0.01) (Figure 2B). By contrast, renin mRNA expression was moderately decreased but not significantly in the nonclipped kidney (2K1C: 1738 ± 342 copies/ng RNA, n.s.) (Figure 2B). Interestingly, neither angiotensinogen mRNA expression, which is the sole substrate of renin (Control: 369 ± 26 copies/ng RNA vs. 2K1C: 416 ± 47 copies/ng RNA, n.s.) (Figure 2C), nor ACE expression, that converts Ang I to Ang II (Control: 1414 ± 136 copies/ng RNA vs. 2K1C: 982 ± 207 copies/ng RNA, n.s.) (Figure 2D), was significantly altered in the clipped and the nonclipped kidney. Increased renin mRNA expression in the clipped kidneys was associated with a significant increase in the circulating Ang II level in response to the development of 2K1C hypertension in WT mice at the end of experiment (Control: 35.8 ± 6.5 fmol/mL vs. 2K1C: 115.4 ± 17.8 fmol/mL, *p* < 0.01) (Figure 2E). Renal cortical Ang II levels were significantly increased not only in the clipped left kidney (Control: 225.4 ± 23.5 pg/g kidney wt. vs. 2K1C: 405.7 ± 46.2 pg/g kidney wt., *p* < 0.01), but also in the nonclipped right kidney (2K1C: 489.8 ± 59.13 pg/g kidney wt., *p* < 0.01) (Figure 2F).

### 2.5. The Development of 2K1C Goldblatt Hypertension Was Completely Blocked in Whole-Body Agtr1a^−/−^ Mice

Compared with WT mice, there were no significant time-dependent increases in systolic, diastolic, and mean blood pressure developed in adult male *Agtr1a*^−/−^ mice with the clipping of the left kidney. Specifically, systolic blood pressure remained at the baseline level throughout the experiment (WT+2K1C: 149 ± 3 mmHg vs. *Agtr1a*^−/−^+2K1C: 91 ± 2 mmHg; *p* < 0.01) (Figure 3A). The lack of blood pressure responses was demonstrated despite that intratubular renin mRNA expression in the cortex increased significantly in both clipped (Control: 10,195 ± 1352 copies/ng RNA vs. 2K1C: 17,378 ± 3103 copies/ng RNA, *p* < 0.01) and nonclipped kidneys (2K1C: 19,620 ± 2844 copies/ng RNA, *p* < 0.01 vs. control) in *Agtr1a*^−/−^ mice during the development of 2K1C hypertension (Figure 3B). However, angiotensinogen mRNAs (Figure 3C) and ACE mRNAs (Figure 3D) were not significantly altered in the clipped and nonclipped kidneys. Ang II levels were also significantly increased in the plasma (Control: 77.0 ± 15.0 fmol/mL vs. 2K1C: 145.3 ± 25.7 fmol/mL, *p* < 0.01) (Figure 3E), clipped (Control: 188.4 ± 19.6 pg/g kidney wt., vs. 2K1C: 507.7 ± 21.4 pg/g kidney wt., *p* < 0.01) (Figure 3F), and nonclipped kidneys (2K1C: 385.8 ± 39.3 pg/g kidney wt., Figure 3F; *p* < 0.01 vs. control), likely due to the lack of AT_1a_ receptors which bind circulating and tissue Ang II [36,37,38,39,40].

### 2.6. The Development of 2K1C Goldblatt Hypertension Was Significantly Attenuated in Proximal Tubule-Specific PT-Agtr1a^−/−^ Mice

Compared with WT mice (Figure 2) as well as whole-body *Agtr1a*^−/−^ mice (Figure 3), proximal tubule-specific PT-*Agtr1a*^−/−^ mice showed very surprising blood pressure and circulating and intratubular RAS responses to the development of 2K1C Goldblatt hypertension (Figure 4). No significant time-dependent increases in systolic, diastolic, and mean blood pressure were observed in adult male PT-*Agtr1a*^−/−^ mice in response to placing the same size of silver clip (0.12-mm) around the left kidney (Figure 4A). More specifically, systolic blood pressure was increased by less than 5 to 10 mmHg throughout the experimental period, which was not significant from time-controls (Control: 103 ± 4 mmHg vs. 2K1C: 109 ± 5 mmHg; n.s.) (Figure 4A). The failure of the development of 2K1C hypertension in PT-*Agtr1a*^−/−^ mice was associated with the lack of significant increases in the expression of renin mRNAs (Control: 2402 ± 134 copies/ng RNA vs. 2K1C: 2517 ± 305 copies/ng RNA, n.s.) (Figure 4B), angiotensinogen mRNAs (Control: 616 ± 47 copies/ng RNA vs. 2K1C: 474 ± 134 copies/ng RNA, n.s.) (Figure 4C), ACE mRNAs (Control: 1740 ± 103 copies/ng RNA vs. 2K1C: 1495 ± 96 copies/ng RNA, n.s.) (Figure 4D), and plasma Ang II (Control: 28.4 ± 4.5 fmol/mL vs. 2K1C: 37.4 ± 6.5 fmol/mL, n.s.) (Figure 4E). However, intratubular Ang II levels in the cortex were significantly increased in both clipped (Control: 225.2 ± 13.7 pg/g kidney wt., vs. 2K1C: 466.6 ± 48.9 pg/g kidney wt., *p* < 0.01) (Figure 4E), and nonclipped kidneys (2K1C: 462.1 ± 26.1 pg/g kidney wt., Figure 4F; *p* < 0.01 vs. control) during the development of 2K1C Goldblatt hypertension likely due to the decreases in AT_1_ (AT_1a_) receptor occupancy in the proximal tubules [36,37,38,39,40].

### 2.7. The Development of 2K1C Goldblatt Hypertension Was Significantly Attenuated in Proximal Tubule-Specific PT-Nhe3^−/−^ Mice

Since NHE3, the most important Na^+^ transporter in the proximal tubules, is the key downstream target of the activation of AT_1_ (AT_1a_) receptors by intratubular Ang II, we hypothesized that deletion of NHE3 selectively in the proximal tubules also attenuates the development of 2K1C Goldblatt hypertension in PT-*Nhe3*^−/−^ mice. Indeed, we demonstrated phenotypic blood pressure responses in PT-*Nhe3*^−/−^ mice (Figure 5A) that were very similar to PT-*Agtr1a*^−/−^ mice during the development of 2K1C Goldblatt hypertension (Figure 4A). No significant time-dependent increases in systolic, diastolic, and mean blood pressure were observed in adult male PT-*Nhe3*^−/−^ mice in response to placing a similar silver clip (0.12-mm) around the left kidney (Figure 5A,B). More specifically, systolic blood pressure was again increased by less than 5 to 10 mmHg throughout the 4-week experimental period, but they were not significant from PT-*Nhe3*^−/−^ time-controls (Control: 104 ± 3 mmHg vs. 2K1C: 110 ± 3 mmHg; n.s.) (Figure 5A,B). Further additional experiments show that the development of Ang II-induced hypertension via systemic Ang II infusion was also significantly attenuated by half in adult male PT-*Nhe3*^−/−^ mice (WT + Ang II: 157 ± 5 mmHg vs. PT-*Nhe3*^−/−^ + Ang II: 129 ± 3 mmHg; *p* < 0.01) (Figure 5C), suggesting a similar contribution of systemic vs. proximal tubule Ang II/AT_1a_ receptor systems in the development of Ang II-induced hypertension [32,33,34,35,41,42].

### 2.8. Deletion of AT_1_ (AT_1a_) Receptors or NHE3 Selectively in the Proximal Tubules Promotes the Natriuretic Response in PT-Agtr1a^−/−^ or PT-Nhe3^−/−^ Mice during the Development of 2K1C Goldblatt Hypertension 

Deletion of AT_1a_ receptors selectively in the proximal tubules not only inhibited the renin expression in the clipped kidney, but also induced marked natriuretic responses in PT*-Agtr1a^−/−^ or* PT*-Nhe3^−/−^* mice. Figure 6 compares the natriuretic responses in WT, whole-body *Agtr1a^−/−^,* and PT*-Agtr1a^−/−^ or* PT*-Nhe3^−/−^* mice in response to the induction of 2K1C Goldblatt hypertension. 24 h urinary sodium excretion in whole-body *Agtr1a^−/−^,* PT-*Agtr1a*^−/−^ and PT*-Nhe3^−/−^* mice increased more than ~18% (*p* < 0.05), ~63% (*p* < 0.01), and ~49% (*p* < 0.01), respectively, compared with WT mice with 2K1C Goldblatt hypertension.

### 2.9. Deletion of AT_1_ (AT_1a_) Receptors Selectively in the Proximal Tubules Attenuates Tubulointerstitial Injury in PT-Agtr1a^−/−^ Mice during the Development of 2K1C Goldblatt Hypertension

Angiotensin II-induced glomerular and tubulointerstitial injury is well recognized in animal models with systemic Ang II infusion [32,33,34,43,44], but whether 2K1C Goldblatt hypertension induces kidney injury in the clipped and nonclipped kidneys and what roles of AT_1_ (AT_1a_) receptors in the proximal tubules have not been studied previously. Figure 7 shows and compares Masson’s Trichome kidney cortical staining in the clipped and nonclipped kidneys of WT and PT-*Agtr1a*^−/−^ mice. Masson’s trichrome staining is widely used to study kidney pathologies especially glomerular and tubulointerstitial fibrosis with blue showing tissue collagen or fibrotic tissues. Overall, there was minimal glomerular and tubulointerstitial fibrosis in the nonclipped right kidneys in both WT and PT-*Agtr1a*^−/−^ mice in response to the development of 2K1C hypertension (Figure 7A vs. Figure 7D). In the clipped left kidneys, however, intensive tubulointerstitial fibrotic responses were observed in WT mice with 2K1C hypertension without significant glomerular injury (Figure 7B). By contrast, tubulointerstitial injury as observed in WT mice with 2K1C hypertension was largely attenuated in the clipped kidneys of PT-*Agtr1a*^−/−^ mice (Figure 7E). Further experiments shows that in WT mice with Ang II-induced hypertension, systemic infusion of a pressor dose of Ang II (1.5 mg/kg/day, i.p.) for 2 weeks induced not only marked tubulointerstitial but also glomerular injury (Figure 7C). Interestingly, these glomerular and tubulointerstitial injuries observed in Ang II-induced hypertension in WT mice were almost completely attenuated in Ang II-infused PT-*Agtr1a*^−/−^ mice (Figure 7F).

## 3. Discussion

The present study used novel mouse models with genetic deletion of the AT_1a_ receptor or Na^+^/H^+^ exchanger 3 (NHE3) selectively in the proximal tubules of the kidney to test the hypothesis for the first time that intratubular AT_1a_ receptors, via its downstream target Na^+^ transporter NHE3, in the proximal tubules are required for the development and maintenance of 2K1C Goldblatt hypertension in mice. We demonstrated that unilateral clipping of the left kidney induced time-dependent increases in telemetry systolic, diastolic, and mean arterial blood pressure in WT mice with a maximal pressor response by Week 3. As expected, 2K1C hypertension in WT mice was associated with increases in renin mRNA expression in the clipped left kidney, and circulating and kidney cortical Ang II levels, accompanied by significant tubulointerstitial injury in the clipped kidney. The development of 2K1C Goldblatt hypertension and kidney injury in WT mice was clearly mediated by AT_1a_ receptors, because whole-body deletion of AT_1a_ receptors in all tissues completely blocked the development of 2K1C hypertension in *Agtr1a*^−/−^ mice. A critical role for Ang II via activating AT_1_ (AT_1a_) receptors to induce 2K1C Goldblatt hypertension has been reported previously in rats [10,11,12,13,14], mice [15,16,26,27,28], and humans with secondary hypertension due to renal arterial stenosis that significantly decreases renal blood flow to the kidney [3,8,9]. Our findings derived from WT and *Agtr1a*^−/−^ mice with whole-body deletion of AT_1a_ receptors are largely consistent with the above-mentioned studies. However, this was not the focus of the present study in that we specifically tested the hypothesis for the 1st time that deletion of AT_1a_ receptors or its downstream target NHE3 alone selectively in the proximal tubules of the kidney was enough to attenuate or prevent the development of 2K1C Goldblatt hypertension, thus providing the new evidence for a critical role of proximal tubule AT_1a_ receptors and NHE3 in the pathogenesis of 2K1C hypertension.

Most, if not all, previous animal studies have investigated the mechanisms underlying the development of 2K1C Goldblatt hypertension using Ang II receptor blockers targeting AT_1_ receptors in all tissues [10,12,13,17,22,45], AT_1_ (AT_1a_) receptor antisense oligonucleotides [14,21], or a mutant mouse model with whole-body deletion of AT_1a_ receptors, *Agtr1a*^−/−^ [15,16]. These studies have been instrumental in establishing an important role of AT_1_ (AT_1a_) receptors in the pathogenesis of 2K1C Goldblatt hypertension in animal models. However, we and others have previously shown that AT_1_ (AT_1a_) receptors are widely expressed and localized in virtually all Ang II-targeting tissues or cells, ranging from the brain, heart, blood vessels, kidney, and adrenal glands that contribute to maintenance of basal blood pressure homeostasis and the development of hypertension [43,44,46,47,48,49]. Indeed, even in the kidney alone, AT_1_ (AT_1a_) receptors are expressed and localized in all renal cells, including mesangial cells and podocytes, proximal tubules, Loop of Henle, distal tubules, collecting ducts, as well as renomedullary interstitial cells [43,44,46,47,48,49,50,51,52]. Induction of 2K1C Goldblatt hypertension not only increases the expression of the RAS and Ang II generation in the clipped kidney, but also activates the circulating RAS due to the release of active renin from the clipped kidney and spills into the circulation. Although both circulating and kidney RAS contribute to the pathophysiology of 2K1C hypertension, the kidney RAS perhaps plays a more critical role in our views. This is because the development of 2K1C hypertension in animal models or human subjects primarily originates from renal artery stenosis that activates the kidney RAS first with increases in renin mRNA expression and active renin release into the clipped kidney and the circulation. Indeed, the activation of the circulating RAS is secondary to the activation of the kidney RAS during the development of 2K1C hypertension. Thus, the whole-body treatment with ARBs or *Agtr1a*^−/−^ mice would not be able to determine the contributions of the kidney versus circulating RAS, especially the AT_1a_ receptor, to the development of 2K1C hypertension.

Whether AT_1_ (AT_1a_) receptors and the downstream target NHE3 in the proximal tubules contributes to the development of 2K1C Goldblatt hypertension has never been investigated previously. The mutant mouse models used for testing this hypothesis in the present study are novel and clearly superior to those previous studies using ARBs targeting AT_1_ receptors in all tissues [10,12,13,17,22,45] or the mouse model with whole-body deletion of AT_1a_ receptors [15,16]. Specifically, use of PT-*Agtr1a*^−/−^ mice allowed us to determine whether AT_1_ (AT_1a_) receptors in the proximal tubules are required for the development of 2K1C Goldblatt hypertension, whereas use of digital droplet PCR to measure the expression of major components of intratubular RAS and PT-*Nhe3*^−/−^ mice would help us determine why deletion of proximal tubule AT_1a_ receptors attenuates or prevents the development of 2K1C Goldblatt hypertension. Recently, we have shown that deletion of AT_1a_ receptors or NHE3 selectively in the proximal tubules significantly attenuated Ang II-induced hypertension with systemic Ang II infusion, which strongly suggests that the development of 2K1C hypertension may also be attenuated in PT-*Agtr1a*^−/−^ or PT-*Nhe3*^−/−^ mice [32,33,34,35]. The data from the present study unequivocally showed that compared with WT as well as whole-body *Agtr1a*^−/−^ mice, induction of 2K1C hypertension by clipping the left renal artery did not significantly increase the expression of renin, angiotensinogen, and ACE mRNAs in the clipped kidney of PT-*Agtr1a*^−/−^ mice. The circulating Ang II levels in PT-*Agtr1a*^−/−^ mice remained at about one-third of those in WT mice, suggesting that the expression of renin mRNAs in the kidney was not activated, and active renin was not released from the kidney into the circulation in PT-*Agtr1a*^−/−^ mice during the development of 2K1C hypertension. Although Ang II was moderately increased in the clipped and nonclipped kidneys in PT-*Agtr1a*^−/−^ mice, the increased kidney Ang II in PT-*Agtr1a*^−/−^ mice may be explained in part by the lack of AT_1a_ receptor occupancy in the proximal tubules due to the deletion of AT_1a_ receptors. Furthermore, the deletion of AT_1a_ receptors selectively in the proximal tubules will make intratubular Ang II ineffective from stimulating NHE3 and other Na^+^ transporters and increasing proximal tubule Na^+^ reabsorption in PT-*Agtr1a*^−/−^ mice. This interpretation is supported by further experiments in the present study that deletion of NHE3 selectively in the proximal tubules, a major downstream target of the activation of the intratubular RAS in the proximal tubules, also attenuated the development of 2K1C hypertension in PT-*Nhe3*^−/−^ mice. Thus, our results strongly support a critical role of proximal tubule AT_1a_ receptors and NHE3 in the development of 2K1C Goldblatt hypertension.

How deletion of proximal tubule AT_1a_ receptors or its downstream target NHE3 in the kidney attenuates the development of 2K1C Goldblatt hypertension remains poorly understood. However, we can reasonably speculate that clipping one of the kidney arteries markedly decreases renal blood flow and glomerular filtration in the clipped kidney, which will induce renin expression and release into the cortical tubular interstitium in the clipped kidney, where it activates intratubular RAS in the proximal tubules. The proximal tubules not only express and accumulate or take up circulating, live-derived and intratubular angiotensinogen [53,54,55,56], but also express abundant ACE and AT_1_ (AT_1a_) receptors [43,44,46,47,48,49,50,51,52]. We hypothesized that 2K1C-induced Ang II formation occurs primarily in the proximal tubules, which directly stimulates AT_1_ (AT_1a_) receptors on apical and basolateral membranes to increase NHE3 expression and proximal tubule Na^+^ reabsorption [32,33,34,35,57,58,59,60]. If this hypothesis is correct, deletion of AT_1a_ receptors or NHE3 selectively in the proximal tubules is expected to attenuate the development of 2K1C Goldblatt hypertension. Indeed, this is supported by our recent studies showing that proximal tubule-specific deletion of AT_1a_ receptors or NHE3 in the kidney not only lowers basal blood pressure by inhibiting proximal tubule Na^+^ reabsorption and promoting the pressure-natriuresis response, but also markedly attenuated Ang II-induced hypertension in PT-*Agtr1a*^−/−^ or PT-*Nhe3*^−/−^ mice [32,33,34,35,57,58].

The present study also demonstrated that deletion of AT_1a_ receptors selectively in the proximal tubules also attenuated 2K1C hypertension-induced renal cortical tubulointerstitial injury. The mechanisms underlying the renal protective effects of AT_1a_ receptor deletion in the proximal tubules remain to be further determined. Previous studies have shown that systemic infusion of Ang II or induction of 2K1C hypertension induced marked glomerular and tubulointerstitial injury or fibrotic responses associated with marked increased expression of growth factors, cytokines, and chemokines in the kidney [4,23,25,35,45,61,62]. AT_1a_ receptors in the glomeruli and proximal tubules undoubtedly play the important roles as demonstrated in studies using ARBs and whole-body *Agtr1a*^−/−^ mice. Further studies are necessary to uncover important signaling mechanisms by which deletion of AT_1a_ receptors in the proximal tubules attenuates tubulointerstitial injury induced by 2K1C Goldblatt hypertension.

In summary, the present study represents the first to use novel mutant mouse models with proximal tubule-specific deletion of AT_1a_ receptors, PT-*Agtr1a*^−/−^, or NHE3, PT*-Nhe3^−/−^*, to directly test the hypothesis that the AT_1a_ receptor and its downstream target Na^+^ transporter NHE3 in the proximal tubules are necessary for the development of 2K1C Goldblatt hypertension. This conclusion is based on our rigorous experimental design, mechanistic tissue- or cell-specific approaches, and careful comparisons of the development of 2K1C Goldblatt hypertension between four different mouse models including WT, whole-body *Agtr1a*^−/−^, and proximal tubule-selective PT-*Agtr1a*^−/−^ and PT*-Nhe3^−/−^* mice. The development of 2K1C hypertension in WT mice is highly expected, but the findings that the development of 2K1C hypertension was almost completely prevented in PT-*Agtr1a*^−/−^ or PT*-Nhe3^−/−^* mice are very surprising. Our data strongly suggest that the development of 2K1C Goldblatt hypertension is not solely dependent on increased renin expression and renin release from the clipped kidney into the circulation, where Ang II induces 2K1C hypertension by activating AT_1a_ receptors in other target tissues [10,12,13,15,16,17,22,45]. More importantly, however, our data support a more prominent role of the intratubular renin/ACE/Ang II/AT_1a_/NHE3 axis in the proximal tubules in the development of 2K1C hypertension. Indeed, deletion of AT_1a_ receptors or NHE3 in the proximal tubules inhibits proximal tubule Na^+^ reabsorption and increases the Na^+^ delivery from the proximal tubules into downstream macula densa cells, which inhibits renin expression and release in the clipped kidney as a feedback response. Further, deletion of AT_1a_ receptors or NHE3 in the proximal tubules effectively block a key pathway in the development and sustenance of 2K1C hypertension, i.e., increased circulating and intratubular Ang II-stimulated proximal tubule Na^+^ reabsorption and induce salt retention by impairing the pressure-natriuresis response [32,33,34,35,59,60]. However, one important limitation of the present study is that it does not completely exclude the contributions of AT_1a_ receptors or NHE3 in other tissues, such as blood vessels, brain, heart, or adrenal glands, to the development of 2K1C Goldblatt hypertension. The use of novel mutant mouse models with deletion of AT_1a_ receptors or NHE3 selectively in any one of these Ang II-targeted tissues may be necessary for future studies to determine their specific contributions to the development of 2K1C Goldblatt hypertension.

## 4. Materials and Methods

For transparency, the authors will make all methods and materials including the generating and genotyping protocols for mutant mice with global or proximal tubule-specific deletion of AT_1a_ receptors or NHE3, the experimental design to induce 2K1C Goldblatt hypertension, raw telemetry and tail-cuff blood pressure data, the measurement of whole-kidney glomerular filtration rate (GFR) using fluorescein-labeled FITC-Sinistrin, etc., available to other researchers, as requested.

### 4.1. Animals

To mechanistically test the hypothesis, we used adult male WT and mutant mice with whole-body deletion of the AT_1a_ receptor (*Agtr1a*^−/−^), proximal tubule-specific deletion of the AT_1a_ receptor (PT-*Agtr1a*^−/−^), and proximal tubule-specific deletion of NHE3 (PT-*Nhe3*^−/−^) in the present study (Figure 8A), as we described previously [36,37,38] and recently [32,33,34,35]. Unless specified elsewhere, WT mice will be *Agtr1a^+/+^*, iL-*Sglt2-Cre^+/-^*, *Nhe3^fl/fl^*, or C57BL/6J, as appropriate, because all strains were generated and back-crossed on the C57BL/6J genetic background [32,33,34,35,36,37,38]. The generation and genotyping of whole-body *Agtr1a*^−/−^ [36,37,38], and proximal tubule-specific PT-*Agtr1a*^−/−^ [34,35] or PT-*Nhe3*^−/−^ mice [32,33], have been reported by us recently. The use of adult male (>16-week-old) WT, *Agtr1a*^−/−^, PT-*Agtr1a*^−/−^, and PT*-Nhe3^−/−^* mice were approved by the Institutional Animal Care and Use Committee of Tulane University School of Medicine. All animal experiments as described in the present study were performed at Tulane Hypertension and Renal Center of Excellence and Department of Physiology at Tulane University School of Medicine.

### 4.2. Induction of 2K1C Goldblatt Hypertension in WT, Global Agtr1a^−/−^, PT-Agtr1a^−/−^ and PT-Nhe3^−/−^ Mice

Two groups (n = 12–15, per group) of adult male WT, whole-body *Agtr1a*^−/−^, proximal tubule-specific PT-*Agtr1a*^−/−^, or PT-*Nhe3*^−/−^ were studied. Briefly, mice were anesthetized with sodium pentobarbital (Oak Pharmaceuticals, Lake Forest, IL, USA; 50 mg/kg, i.p.) and under aseptic conditions, prepared for either sham surgery as time controls or for the induction of 2K1C Goldblatt hypertension as described previously by others [13,15,16,25]. The left renal artery was exposed via a left flank incision and isolated carefully. A silver clip with an internal diameter of 0.12 mm was placed around the left artery for 4 weeks to reduce whole-kidney blood flow and GFR in the clipped kidney by ~50%, based on preliminary studies using 0.05, 0.12, or 0.2 mm silver clips, respectively (Figure 8B). The animals were allowed to recover for 4 weeks of the experimental period.

### 4.3. Induction of Ang II-Induced Hypertension in WT, Global Agtr1a^−/−^, PT-Agtr1a^−/−^ and PT-Nhe3^−/−^ Mice

For direct comparisons between 2K1C and Ang II-induced hypertension, two groups (n = 8–12) of adult male WT, global *Agtr1a^−/−^, PT-Agtr1a^−/−^ and PT-Nhe3^−/−^* mice were infused with a pressor dose of Ang II (1.5 mg/kg/day, i.p.) for 14 days using an osmotic minipump implanted intraperitoneally (Alzet, Cupertino, CA, USA; Model 2002), as we described recently [32,33,34,35,36,37,38]. Systolic, diastolic, and mean arterial blood pressure were determined as described above for 2K1C Goldblatt hypertension.

### 4.4. Implantation of Telemetry Probes for Blood Pressure Measurement in Wild-Type, Global Agtr1a^−/−^, PT-Agtr1a^−/−^, and PT-Nhe3^−/−^ Mice

Basal systolic, diastolic and mean arterial blood pressure and their responses to sham control or the induction of 2K1C Goldblatt hypertension were measured using both implanted telemetry (Data Science, International, St. Paul, MN, USA) and noninvasive tail-cuff technique (Visitech, Apex, NC, USA), as we described [32,33,34,35,36,37,38]. 50% of animals from each group were subject to the implantation of a telemetry probe, whereas the remaining animals from each group were subject to the tail-cuff approach for blood pressure measurements. For telemetry implantation, the animals were anesthetized with sodium pentobarbital (50 mg/kg, i.p.), and the probe was implanted into the left carotid artery under aseptic conditions. The catheter was secured in place with a small drop of vet bond adhesive and further secured by a 4.0 surgical silk, if necessary, whereas the telemetry unit was loosely secured to the abdominal wall muscle with 4.0 surgical silk suture [32,33,34,35]. All animals were allowed to recover from anesthesia and surgery for the duration of the experiment. One week after telemetry implantation, systolic, diastolic, and mean arterial blood pressure were determined under basal conditions. This was followed by either sham surgery or induction of 2K1C Goldblatt hypertension during the following week, and systolic, diastolic, and mean arterial blood pressure were determined daily during the 1st week and weekly for next 3 weeks, as we described recently [32,33,34,35,36,37,38].

### 4.5. Measurement of Whole-Kidney Glomerular Filtration Rate (GFR) Using Fluorescein Labeled FITC-Sinistrin in WT, Global Agtr1a^−/−^, and PT-Agtr1a^−/−^ Mice

To determine the kidney function at the end of experiment, adult male WT, global *Agtr1a*^−/−^, and PT-*Agtr1a*^−/−^ mice were anesthetized with sodium pentobarbital (50 mg/kg, i.p.) and surgically prepared for cannulation of the left jugular vein for a bolus injection of fluorescein labeled FITC-sinistrin (10 mg/kg, i.v., Medibeacon, St. Louis, MO, USA), as we described recently [32,33,34,35]. FITC-Sinistrin levels in skin microcirculation was continuously measured using a battery-powered transdermal GFR monitor, which was attached directly to a shaved and cleaned left flank skin, for up to 3 h to determine the half-time clearance rate [32,33,34,63,64]. Whole-kidney GFR was calculated from the half-life curves of FITC-Sinistrin clearance from the circulation using the two-compartment model and the mouse specific conversion factor, as previously described [32,33,34,63,64].

### 4.6. Glomerular and Tubulointerstitial Histology in Wild-Type and Proximal Tubule-Specific PT-Agtr1a^−/−^ Mice

To determine the roles of AT_1_ (AT_1a_) receptors in the proximal tubules in 2K1C hypertension-induced kidney injury, we compared glomerular and renal cortical tubulointerstitial injuries during the development of 2K1C Goldblatt hypertension. The kidneys were collected from WT and PT-*Agtr1a*^−/−^ mice at the end of experiment and processed for Masson’s Trichome staining to visualize glomerular and tubulointerstitial injury, as we recently described [32,33,34,35].

### 4.7. Digital Droplet PCR Analysis of the Expression of Renin, Angiotensinogen, and Angiotensin-Converting Enzyme mRNAs in the Clipped and Nonclipped Kidneys of WT, Global Agtr1a^−/−^, and PT-Agtr1a^−/−^ Mice

To determine the expression of major components of the RAS in the kidney in response to the development of 2K1C Goldblatt hypertension, kidney cortical samples were obtained from WT, global *Agtr1a*^−/−^, and PT-*Agtr1a*^−/−^ mice at the end of the experiment. Digital droplet PCR (ddPCR) was used to determine the expressed copies of angiotensinogen, renin, angiotensin-converting enzyme (ACE) mRNAs, respectively, as described [65,66]. Briefly, total RNA was isolated from all kidney cortical samples using a commercial RNA isolation kit (Qiagen, Hilden, Germany), RNA concentrations quantified using Nanodrop 2000 (Thermo Scientific, Waltham, MA, USA), whereas ddPCR was performed using a Bio-Rad ddPCR system. All primers, probes, and reagents for the One-step RT-ddPCR system were custom designed and provided by Bio-Rad to generate cDNA and quantify gene expression, and the copy number of each target gene was analyzed using the QX200 droplet reader and the QuantaSoft analysis Pro 1.0 software (Bio-Rad, Hercules, CA, USA). Data are expressed as copy numbers of each target gene in 1 ng total RNA [65,66].

### 4.8. Measurement of Plasma and Kidney Ang II Levels in Wild-Type, Global Agtr1a^−/−^, and PT-Agtr1a^−/−^ Mice

At the end of experiment, all mice were decapitated, and trunk blood samples were collected in a tube containing an inhibitor cocktail to prevent the generation and degradation of Ang II, as we previously described [32,33,34,35,36,37,38]. One kidney was collected from each mouse, sliced, and minced into an extraction buffer containing an inhibitor cocktail to prevent Ang II generation and degradation [32,33,34,35,36,37,38]. Ang II peptides were extracted using a phenyl-bonded solid-phase peptide extraction column (Elut-C18, Varian, Palo Alto, CA, USA), vacuum-dried overnight, and reconstituted in an Ang II assay buffer before being measured using a sensitive ELISA kit (Bachem, Torrance, CA, USA) [32,33,34,35,36,37,38].

### 4.9. Measurement of Blood Volume, Hematocrit and 24 h Urinary Na^+^, K^+^, and Cl^−^ Excretion

At the end of experiment, all mice were decapitated, and trunk blood samples were collected, and blood volume measured gravimetrically, whereas hematocrit (Hct) was measured using the microcapillary method [32,33,34,35]. To measure 24 h urinary Na^+^, K^+^ and Cl^−^ excretion, all mice were housed individually in a specialized mouse metabolic cage with free access to H_2_O for drinking and a standard rodent chow. Urine samples were collected at baselines and weekly during the development of 2K1C hypertension for 24 h into a plastic tube containing baby oil to prevent samples’ evaporation. Na^+^, K^+^ and Cl^−^ concentrations were determined using a Nova 13 Electrolyte Analyzer (Nova Biomedical, Waltham, MA, USA), as we described previously [32,33,34,35].

### 4.10. Statistical Analysis

All results are presented as mean ± SEM. The differences under basal conditions or in response to the development of 2K1C Goldblatt hypertension between different groups of WT, global *Agtr1a*^−/−^**,** PT-*Agtr1a*^−/−^, and PT-*Nhe3*^−/−^ mice including systolic, diastolic and mean arterial pressure, whole-kidney GFR, 24 h urinary Na^+^ excretion, the expression of major components of the RAS in the kidney were first analyzed using one-way ANOVA, followed by Student’s unpaired *t* test if a significant response between groups of mice was detected. The significance of differences between different groups were set *p* < 0.05.

## Figures and Tables

**Figure 1 ijms-23-15798-f001:**
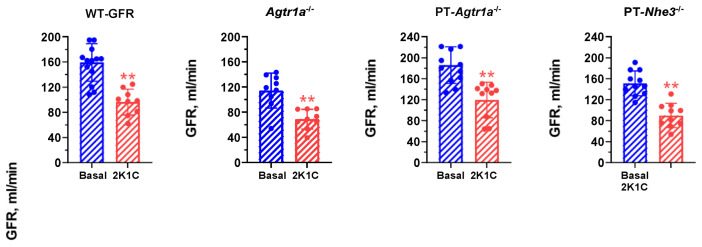
Induction of 2K1C Goldblatt hypertension with 0.12 mm silver clip on the left kidney significantly decreases glomerular filtration rate (GFR), as measured using FITC-Sinistrin, equally by >40% in WT, global *Agtr1a*^−/−^, proximal tubule-specific PT-*Agtr1a*^−/−^ or PT-*Nhe3*^−/−^ mice. N = 9–15 per strain, per group. ** *p* < 0.01 vs. Basal.

**Figure 2 ijms-23-15798-f002:**
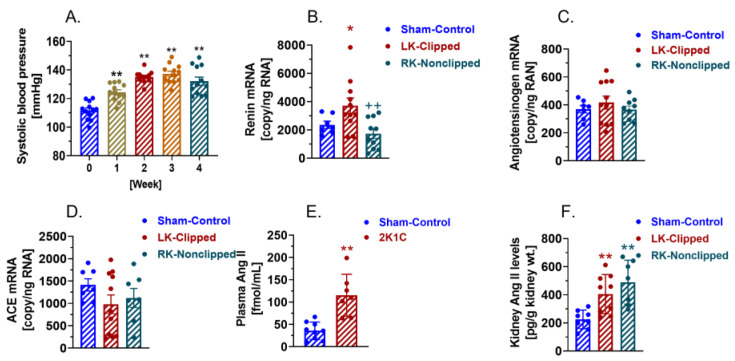
The development of 2K1C Goldblatt hypertension in WT mice with the activation of circulating and intratubular renin-angiotensin system. (**A**), the time-dependent increases in systolic blood pressure during the development of 2K1C hypertension. (**B**), increased intratubular expression of renin mRNAs in the clipped kidney. (**C**), no changes in intratubular expression of angiotensinogen mRNAs between control, clipped, and nonclipped kidneys. (**D**), no changes in intratubular expression of ACE mRNAs between control, clipped, and nonclipped kidneys. (**E**), Increased circulating Ang II level during the development of 2K1C hypertension. (**F**), increased intratubular Ang II in both clipped and contralateral nonclipped kidney during the development of 2K1C hypertension. Note that Control is defined as sham-operated control mice without clipping the left renal artery, LK as the left clipped kidney, and RK as the unclipped contralateral kidney, respectively. * *p* < 0.05 or ** *p* < 0.01 vs. control or baseline; ^++^
*p* < 0.01 vs. the clipped kidney.

**Figure 3 ijms-23-15798-f003:**
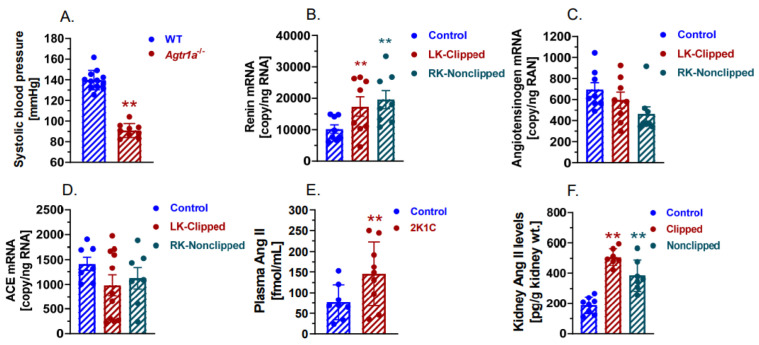
Deletion of the AT_1a_ receptor prevents the development of 2K1C Goldblatt hypertension in *Agtr1a*^−/−^ mice with whole-body AT_1a_ receptor knockout. (**A**), comparison of the maximal blood pressure responses between WT and *Agtr1a*^−/−^ mice during the development of 2K1C hypertension. (**B**), increased intratubular expression of renin mRNAs in both clipped and nonclipped kidneys. (**C**), no changes in intratubular expression of angiotensinogen mRNAs between control, clipped, and nonclipped kidneys. (**D**), no changes in intratubular expression of ACE mRNAs between control, clipped, and nonclipped kidneys. (**E**), Increased circulating Ang II during the development of 2K1C hypertension in *Agtr1a*^−/−^ mice. (**F**), increased intratubular Ang II in both clipped and contralateral nonclipped kidney during the development of 2K1C hypertension in *Agtr1a*^−/−^ mice. Note that Control is defined as sham-operated control mice without clipping the left renal artery, LK as the left clipped kidney, and RK as the unclipped contralateral kidney, respectively. ** *p* < 0.01 vs. control or baseline.

**Figure 4 ijms-23-15798-f004:**
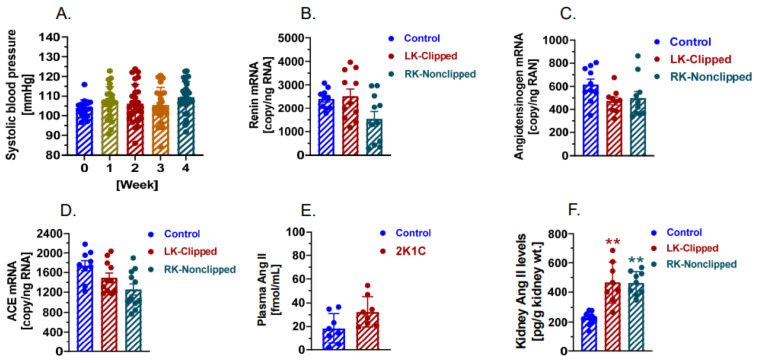
Proximal tubule-specific deletion of AT_1a_ receptors in the kidney significantly attenuates the development of 2K1C Goldblatt hypertension in PT-*Agtr1a*^−/−^ mice. (**A**), the lack of the time-dependent systolic blood pressure responses for 4 weeks during the development of 2K1C hypertension. (**B**), no change in intratubular expression of renin mRNAs in both clipped and nonclipped kidneys. (**C**), no changes in intratubular expression of angiotensinogen mRNAs between control, clipped, and nonclipped kidneys. (**D**), no changes in intratubular expression of ACE mRNAs between control, clipped, and nonclipped kidneys. (**E**), circulating Ang II slightly but not statistically increased during the development of 2K1C hypertension. (**F**), increased intratubular Ang II in both clipped and contralateral nonclipped kidney during the development of 2K1C hypertension likely due to the decreases in AT_1_ (AT_1a_) receptor occupancy in the proximal tubules. ** *p* < 0.01 vs. control kidney.

**Figure 5 ijms-23-15798-f005:**
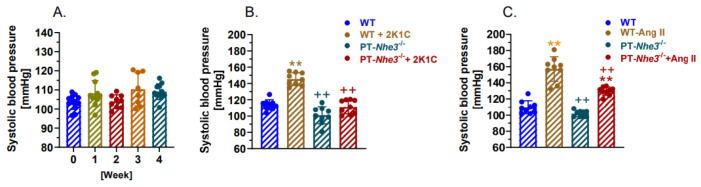
Proximal tubule-specific deletion of the Na^+^/H^+^ exchanger 3 (NHE3), a major downstream target of AT_1a_ receptor activation by intratubular Ang II, in the kidney significantly attenuates the development of 2K1C Goldblatt hypertension in PT-*Nhe3*^−/−^ mice. (**A**), the lack of the time-dependent systolic blood pressure responses for 4 weeks during the development of 2K1C hypertension, a phenotypic response similar to PT-*Agtr1a*^−/−^ mice. (**B**), comparisons of maximal systolic blood pressure responses at Week 3 between wild-type and PT-*Nhe3*^−/−^ mice during the development of 2K1C hypertension. (**C**), comparisons of maximal systolic blood pressure responses to 2-week systemic infusion of a pressor dose of Ang II, 1.5 mg/kg/day, i.p., between wild-type and PT-*Nhe3*^−/−^ mice. ** *p* < 0.01 vs. control or baseline in the same mouse strain. ^++^
*p* < 0.01 vs. wild-type control or during the development of 2K1C Goldblatt hypertension (**B**) or Ang II-induced hypertension (**C**).

**Figure 6 ijms-23-15798-f006:**
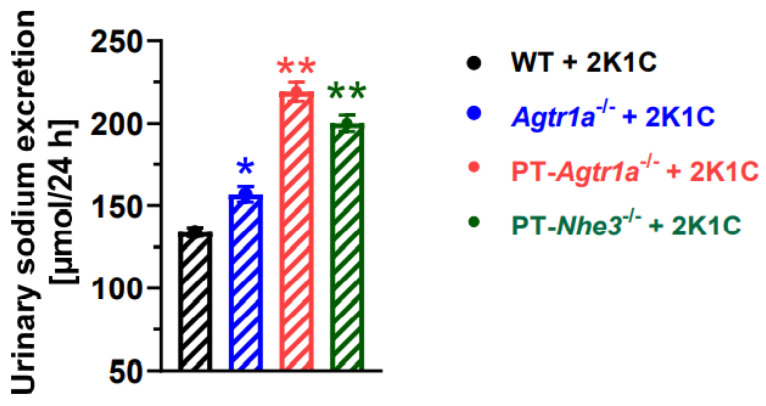
Deletion of AT_1a_ receptors or NHE3 selectively in the proximal tubules of the kidney promotes the natriuretic responses to the induction of 2K1C Goldblatt hypertension in proximal tubule-specific PT-*Agtr1a*^−/−^ or PT-*Nhe3*^−/−^ mice. N = 9–15 per strain, per group. * *p*< 0.05 or ** *p* < 0.01 vs. WT + 2K1C.

**Figure 7 ijms-23-15798-f007:**
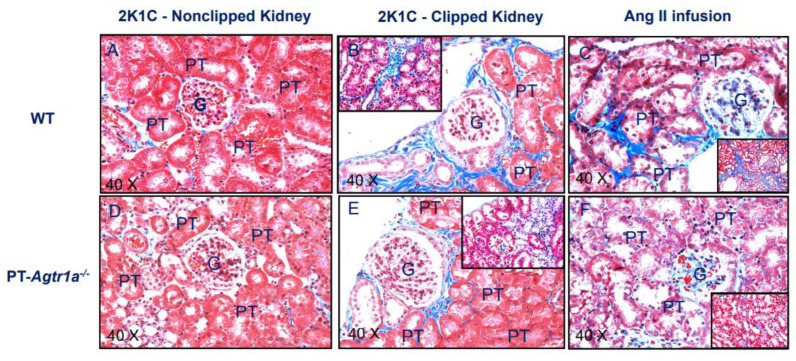
Deletion of AT_1a_ receptors selectively in the proximal tubules of PT-*Agtr1a*^−/−^ mice significantly attenuates the development of tubulointerstitial injury in the renal cortex induced by 2K1C Goldblatt hypertension, as revealed by Masson’s Trichome staining. (**A**), the nonclipped contralateral kidney in a representative WT mouse kidney. (**B**), the development of marked tubulointerstitial injury in a representative WT mouse kidney in response to the development of 2K1C hypertension with blue color staining in the peri-glomerular and tubulointertsitial areas. (**C**), the development of marked tubulointerstitial injury in a representative WT mouse kidney in response to the development of Ang II-induced hypertension with blue color staining in the peri-glomerular and tubulointertsitial areas. (**D**), the nonclipped kidney of a representative PT-*Agtr1a*^−/−^ mouse. (**E**), marked tubulointerstitial injury as developed in WT mice in response to the development of 2K1C hypertension was attenuated in a representative PT-*Agtr1a*^−/−^ mouse kidney. (**F**)**,** marked tubulointerstitial injury as developed in WT mice in response to the development of Ang II-induced hypertension was also markedly attenuated in a representative PT-*Agtr1a*^−/−^ mouse kidney. G, glomerulus; PT, proximal tubule. Insets represent the cortical tubulointerstitial areas.

**Figure 8 ijms-23-15798-f008:**
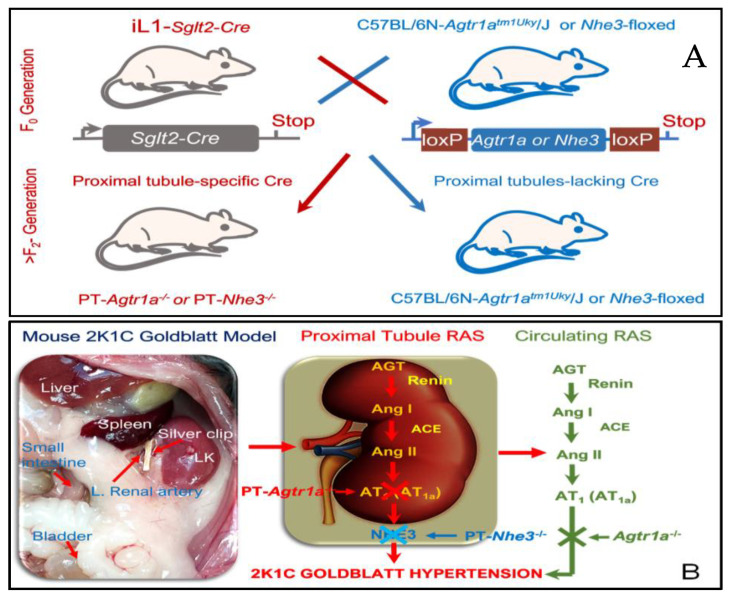
The schematic representation showing the iL-*Sglt2-Cre*/LoxP approach to generate mutant mouse models with proximal tubule-specific deletion of the AT_1a_ receptor or its downstream target protein NHE3 [32,33,34,35] (**A**) and the working hypothesis that deletion of the AT_1a_ receptor or NHE3 selectively in the proximal tubules attenuates or blocks the development of 2K1C Goldblatt hypertension (**B**).

**Table 1 ijms-23-15798-t001:** Basal physiological phenotypes of adult male wild-type (WT), global *Agtr1a*^−/−^, proximal tubule-specific PT-*Agtr1a*^−/−^, and PT*-Nhe3^−/−^* mice.

Parameter	WT Control N = 13	*Agtr1a*^−/−^ Control N = 9	PT-*Agtr1a*^−/−^ Control N = 13	PT-*Nhe3*^−/−^ Control N = 13
Body wt., g	28.6 ± 0.4	29.2 ± 0.7	29.3 ± 0.4	31.8 ± 1.2
Heart wt., g	0.14 ± 0.01	0.14 ± 0.01	0.15 ± 0.01	0.16 ± 0.01
Heart wt./Body wt. ratio	0.50 ± 0.02	0.47 ± 0.01 *	0.52 ± 0.01 ^†^	0.52 ± 0.02 ^†^
Left kidney wt., g	0.18 ± 0.01	0.14 ± 0.01 **	0.17 ± 0.01 ^††^	0.19 ± 0.01 ^††^
Let kidney wt./body wt. ratio	0.62 ± 0.03	0.49 ± 0.02 **	0.59 ± 0.03 ^††^	0.62 ± 0.02 ^††^
Blood volume, mL	0.72 ± 0.04	0.73 ± 0.02	0.76 ± 0.03	0.82 ± 0.05 ^††^
Hematocrit, %	44 ± 0.7	43 ± 0.6	45 ± 0.4	46 ± 0.5
Urine excretion, mL/24 h	1.47 ± 0.10	1.90 ± 0.09 **	1.40 ± 0.03 ^†^	1.33 ± 0.09 ^††^
Na+ excretion, µmol/24 h	156.2 ± 4.8	237.3 ± 10.7 **	222.6 ± 6.9 **	186.1 ± 7.4 *^†^
K+ excretion, µmol/24 h	221.0 ± 7.4	384.8 ± 14.7 **	317.2 ± 14.1 ^†^	264.9 ± 17.4 ^††^
Urine Cl^−^ excretion, µmol/24 h	242.1 ± 11.6	404.2 ± 45.9 **	344.7 ± 25.2 ^†^	260 ± 11.1 ^††^

* *p* < 0.05 or ** *p* < 0.01 vs. WT; ^†^
*p* < 0.05 or ^††^ *p* < 0.01 vs. *Agtr1a*^−/−^.

**Table 2 ijms-23-15798-t002:** Physiological phenotypic responses to the development of 2K1C Goldblatt hypertension in adult male wild-type (WT), global *Agtr1a*^−/−^ and proximal tubule-specific PT-*Agtr1a*^−/−^ or PT-*Nhe3*^−/−^ mice.

Parameter	WT + 2K1C N = 15	*Agtr1a*^−/−^ + 2K1C N = 9	PT-*Agtr1a*^−/−^ + 2K1C N = 15	PT-*Nhe3*^−/−^ + 2K1C N = 12
Body wt., g	27.0 ± 0.5	28.4 ± 0.6	29.1 ± 0.4	28.3 ± 0.6
Heart wt., g	0.14 ± 0.01	0.13 ± 0.01	0.15 ± 0.01	0.13 ± 0.01
Heart wt./Body wt. ratio	0.53 ± 0.02	0.47 ± 0.01 **	0.52 ± 0.01 ^†^	0.47 ± 0.01
Left kidney wt., g	0.14 ± 0.01	0.15 ± 0.02	0.16 ± 0.01	0.15 ± 0.01
Let kidney wt./body wt. ratio	0.54 ± 0.03	0.53 ± 0.01	0.56 ± 0.02	0.57 ± 0.02
Blood volume, mL	0.66 ± 0.04	0.80 ± 0.02 **	0.72 ± 0.03 *^†^	0.71 ± 0.03 *^†^
Hematocrit, %	47 ± 0.5	43 ± 0.4 **	45 ± 0.4	45 ± 0.5
Urine excretion, mL/24 h	0.93 ± 0.04	1.80 ± 0.06 **	1.65 ± 0.04 **	1.54 ± 0.03 **
Na+ excretion, µmol/24 h	134.1 ± 2.6	156.8 ± 4.8 **	219.2 ± 5.7 **^††^	200.3 ± 5.0 **^††^
K+ excretion, µmol/24 h	170.6 ± 3.8	281.9 ± 10.5 **	372.3 ± 11.6 **^††^	312.2 ± 14.1 **
Urine Cl^−^ excretion, µmol/24 h	163.9 ± 3.8	506.0 ± 39.6 **	336.3 ± 9.8 **^††^	303.6 ± 7.6 **^††^

* *p* < 0.05 or ** *p* < 0.01 vs. WT + 2K1C; ^†^
*p* < 0.05 or ^††^ *p* < 0.01 vs. *Agtr1a*^−/−^ + 2K1C.

## Data Availability

The authors will make all methods and materials including the protocols for genotyping mutant mice with proximal tubule-specific deletion of AT_1a_ or NHE3 receptors, surviving and non-surviving mouse surgical protocols, experimental design, and protocols, and all supporting raw data available to other researchers upon requests.

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
