# Peer review of "Genetic Deletion of AT_1a_ Receptor or Na^+^/H^+^ Exchanger 3 Selectively in the Proximal Tubules of the Kidney Attenuates Two-Kidney, One-Clip Goldblatt Hypertension in Mice"

_ijms, 2022, doi:10.3390/ijms232415798_

Round 1
Reviewer 1 Report
This is an interesting study showing the key factors i.e. AT1a receptor and NHE3 localized in proximal tubule, engaged in development of hypertension in mouse 2K1C model. The study design and methods used are elegant, and the results add new knowledge in this field.
In general, the manuscript is well written and easy to follow, but I have a few comments to consider by the authors:
1. The Methods does not comment on the Ang II infusion experiments, but some data from such experiments are presented in Figure 6 and 8, and commented on in the Results (e.g. lines 365-370; 377-381; 411-416) and Discussion section. Some data are identical to those presented earlier in the authors' own work (No. 35)! This should be corrected.
2. Please add an annotation about the statistical symbols to the legend of Table 2
3. Please check the correctness of Fig. 3: the charts are shifted and do not match panels A, B, C etc. The numbers are wrong size and the whole figure is hard to read.
4. In all figures, statistical markers are difficult to distinguish from individual points assigned to individual bars; please make them more visible.
5. Legend of Fig. 8 – the letter "F" should be bold
6. Lines 531-534: please check the correctness of this sentence
7. The conclusion paragraph is a repeat of the previous paragraph; in my opinion this can be shortened.
8. Line 760: the year of publication should appear immediately after the journal name: Zhuo JL et al.,...
Author Response
Authors’ Responses to Reviewer #1:
Reviewer’s Comment #1: The Methods does not comment on the Ang II infusion experiments, but some data from such experiments are presented in Figure 6 and 8, and commented on in the Results (e.g. lines 365-370; 377-381; 411-416) and Discussion section. Some data are identical to those presented earlier in the authors' own work (No. 35)! This should be corrected.
Authors’ Responses: We would like to thank the reviewer for carefully review our manuscript and provide insightful comments and suggestions. However, we would like to clarify with the reviewer that none of the Ang II infusion data (Fig. 6C; Fig. 8C&F) were from previously published studies in Ang II-infused mutant mice with proximal tubule-specific deletion of AT1a receptors, PT-Agtr1a-/-, or with proximal tubule-specific deletion of NHE3 (References #32-35). Although blood pressure and histological responses to Ang II infusion are similar to those previously published studies, the data included in this manuscript were from additional/different experiments. Still, we accept the reviewer’s suggestions to revise the results’ section to make it clear that are not the previously published data.
Reviewer’s Comment #2: Please add an annotation about the statistical symbols to the legend of Table 2.
Authors’ Response: An annotation about the statistical symbols are now added in the revised manuscript.
Reviewer’s Comment #3: Please check the correctness of Fig. 3: the charts are shifted and do not match panels A, B, C etc. The numbers are wrong size and the whole figure is hard to read.
Authors’ Response: Many thanks for pointing out the errors in this figure. I think that these errors were made from the editing processes in the editorial office, not our provided figure 3. We will alert these errors to the editorial/production team.
Reviewer’s Comment #4: In all figures, statistical markers are difficult to distinguish from individual points assigned to individual bars; please make them more visible.
Authors’ Response: Again, we thought that the sizes of all statistical difference markers are proportional to the panels or the figures. The differences are that the journal production team made the figures in the manuscript smaller than they should be.
Reviewer’s Comment #5: Legend of Fig. 8 – the letter "F" should be bold.
Authors’ Response: It has been corrected.
Lines 531-534: please check the correctness of this sentence
Reviewer’s Comment #6: Lines 531-534: please check the correctness of this sentence.
Authors’ Response: This sentence has been corrected as you suggested. Thank you!
Reviewer’s Comment #7: The conclusion paragraph is a repeat of the previous paragraph; in my opinion this can be shortened.
Authors’ Response: The conclusion paragraph has been revised and shortened as the reviewer suggested.
Reviewer’s Comment #8: Line 760: the year of publication should appear immediately after the journal name: Zhuo JL et al.,...
Authors’ Response: The reviewer is correct, and this error has been corrected.
Reviewer 2 Report
General comments:
Li XC et al. investigated a pathophysiological significance of renal proximal tubule (PT) AT1aR and NHE3 in the development of 2K1C Goldblatt hypertension that replicates features of human renal vascular hypertension. In this model, wild type mice exhibited continuous blood pressure elevation with increases in renal mRNA expression of renin and plasma/renal Ang II levels, while renal mRNA expression of angiotensinogen and ACE was not affected. Systemic AT1aR deficiency eliminated the development of 2K1C hypertension even though renal mRNA expression of renin and plasma/renal Ang II levels were significantly increased. Interestingly, PT specific-AT1aR deficiency also eliminated the development of 2K1C hypertension, but renal mRNA expression of renin and plasma Ang II levels were not increased. In addition, PT specific-NHE3 deficiency significantly attenuated the development of 2K1C hypertension and Ang II-induced hypertension. Based on these findings, the authors conclude that PT-Ang II/AT1aR/NHE3 signaling plays a critical role in the development of 2K1C hypertension. The results of this manuscript are derived from three types of genetically modified mice (systemic AT1aR, PT-specific AT1aR and PT-specific NHE3 deficient mice) with established experimental methods, and the manuscript itself is well written. The reviewer has several comments as follows.
Major comments:
1. Just comparing the phenotype between PT-specific AT1aR and PT-specific NHE3 deficient mice cannot explain the causality between proximal tubular AT1aR and NHE3 in the development of 2K1C hypertension. Thus, the authors should not describe “AT1aR/NHE3 signaling” throughout the manuscript. Otherwise, at least renal mRNA expression and/or protein levels of NHE3 should be examined in the PT-AT1aR deficient mice to valid the authors’ conclusions.
2. Would renal and plasma RAS components be affected by PT-specific NHE3 deficiency?
3. In addition to urinary sodium excretion, sodium retention estimated by sodium intake and excretion should be shown in Figure 7.
4. The reviewer could not understand the authors’ explanation regarding the mechanism how PT-specific AT1aR deficiency attenuate the development of 2K1C hypertension. The sentence in page 7, line 500-503 is confusing, because PT-specific AT1aR deficiency should accelerate urinary sodium excretion via inactivation of NHE3. In addition, please further discuss the mechanism how PT-specific AT1aR deficiency attenuate renal renin activation in 2K1C hypertension.
5. Renal mRNA expression of angiotensinogen and ACE in wild type mice was not affected in the development of 2K1C hypertension. These results suggest that circulating RAS rather than renal tissue RAS plays a critical role in this model. Overall, in the pathophysiology of 2K1C hypertension, which is critical? Or both? Please clarify this point.
Minor comments:
1. Methods of blood volume, Hct and urinary sodium excretion measurements are missing. Also, protocol of the Ang II infusion experiment is not described in the Methods section.
2. To compare not only wild type vs genetic modified mice but also control vs 2K1C, it would be better to combine Table 1 with Table 2.
3. Why is heart wt./Body wt. ratio comparable between WT-Control and WT-2K1C mice despite the significant blood pressure elevation in WT-2K1C mice.
4. Definitions of “Control” in Figure 3, 4 and 5 are missing. Are they sham-operated mice?
5. Units of Na, K, Cl excretion in Table 1 would be incorrect.
6. Legends of Figure 4B and 4C are oppositely described.
Author Response
Authors’ Responses to Reviewer #2:
Reviewer #2's major comment #1: Just comparing the phenotype between PT-specific AT1aR and PT-specific NHE3 deficient mice cannot explain the causality between proximal tubular AT1aR and NHE3 in the development of 2K1C hypertension. Thus, the authors should not describe “AT1aR/NHE3 signaling” throughout the manuscript. Otherwise, at least renal mRNA expression and/or protein levels of NHE3 should be examined in the PT-AT1aR deficient mice to valid the authors’ conclusions.
Authors’ responses: We agree with the reviewer’s comments. Although NHE3 is the downstream target of the intratubular Ang II/AT1a receptors in the proximal tubules, Ang II also acts on many other signaling pathways other than NHE3. Accordingly, we accept the reviewer’s critique to change the “AT1a receptor/NHE3 signaling” to the “AT1a receptor/NHE3 axis” to reflect these differences. By the way, we have previously reported that Ang II activates AT1a receptors to increase NHE3 mRNA expression in cultured proximal tubule cells and in rats or mice infused with low pressor doses of Ang II infusion.
Reviewer #2's major comment #2: Would renal and plasma RAS components be affected by PT-specific NHE3 deficiency?
Authors’ responses: In theory, renal and plasma RAS components may be affected by PT-specific NHE3 deficiency. We have previously shown that renal and plasma RAS was significantly upregulated in global NHE3-/- with or without transgenic rescue of the NHE3 gene selectively in small intestines. This is because that whole-body NHE3 deficiency causes severe salt wasting which activates circulating and kidney RAS to compensate for the loss of NHE3 in small intestines and the proximal tubules. However, we have not seen significant upregulation of circulating and kidney RAS to the extent in whole-body NHE3 knockout, as salt is not wasted from small intestines and distal tubule nephron segments or other sodium transporters may partially compensate for the loss of NHE3 in the proximal tubules.
Reviewer #2's major comment #3: In addition to urinary sodium excretion, sodium retention estimated by sodium intake and excretion should be shown in Figure 7.
Authors’ responses: The reviewer’s suggestion may be sound, but in reality, sodium retention or sodium balance as estimated by sodium intake and excretion is very hard to do in mice. We have tried to do these experiments previously, but the data are not reliable for any publications.
Reviewer #2's major comment #4: The reviewer could not understand the authors’ explanation regarding the mechanism how PT-specific AT1aR deficiency attenuate the development of 2K1C hypertension. The sentence in page 7, line 500-503 is confusing, because PT-specific AT1aR deficiency should accelerate urinary sodium excretion via inactivation of NHE3. In addition, please further discuss the mechanism how PT-specific AT1aR deficiency attenuate renal renin activation in 2K1C hypertension.
Authors’ responses: With respect to the authors’ explanations regarding the mechanisms on how PT-specific AT1a receptor deficiency attenuates the development of 2K1C hypertension, the reviewer should check these explanations in detail in the discussion in Lines 493-544. The discussion suggests the following mechanisms or explanations: a) inhibition of renin mRNA expression and release into the circulation from the clipped kidney that prevents systemic and intrarenal Ang II generation; b) inhibition of proximal tubule Na+ reabsorption that leads to increased delivery of Na+ to the JGAs which in turn impairs the tubuloglomerular feedback response in the absence of AT1a receptors in the proximal tubules; c) inhibition of proximal tubule Na+ reabsorption via its downstream target NHE3, which normally contributes to the reabsorption of about 50% of glomerularly filtered Na+ load in the proximal tubules; and d) attenuation of 2K1C-induced kidney injury in the clipped kidneys.
Reviewer #2's major comment #5: Renal mRNA expression of angiotensinogen and ACE in wild type mice was not affected in the development of 2K1C hypertension. These results suggest that circulating RAS rather than renal tissue RAS plays a critical role in this model. Overall, in the pathophysiology of 2K1C hypertension, which is critical? Or both? Please clarify this point.
Authors’ responses: Thanks for the comments and your expert insights. We believe that both circulating and renal tissue RAS contribute to the pathophysiology of 2K1C hypertension, but renal tissue RAS perhaps more critical in our views. This is because the development of 2K1C hypertension in wild-type mice or humans originates from the renal artery stenosis that first activates renal tissue RAS, especially renal renin mRNA expression and active renin release into the clipped kidney and the circulation. The activation of the circulating RAS is secondary to the activation of the renal tissue RAS due to renal artery stenosis. If the renal artery stenosis is corrected, then 2K1C hypertension would not develop in the absence of renal artery stenosis.
Reviewer #2's minor comment #1: Methods of blood volume, Hct and urinary sodium excretion measurements are missing. Also, protocol of the Ang II infusion experiment is not described in the Methods section.
Authors’ responses: Thanks for the comments and suggestions. However, the methods for measurements of blood volume, Hct and urinary sodium excretion have been reported extensively in our previously published studies (Ref #32-35 and many others), and therefore omitted to shorten the length of the manuscript.
Reviewer #2's minor comment #2: To compare not only wild type vs genetic modified mice but also control vs 2K1C, it would be better to combine Table 1 with Table 2.
Authors’ responses: Thanks for the suggestion, however, just if we combine two tables and the table will contain too many data columns and rows, which will significantly cause more confusions and less clarity to the readers.
Reviewer #2's minor comment #3: Why is heart wt./Body wt. ratio comparable between WT-Control and WT-2K1C mice despite the significant blood pressure elevation in WT-2K1C mice.
Authors’ responses: This is a good question from the reviewer. In theory, they should be different with significant cardiac hypertrophy in WT-2K1C, as we and many others showed previously in Ang II-infused hypertension with a significant pressor dose of Ang II. However, dependent on the size of the silver clip used to produce 2K1C hypertension, the degree of hypertension in different studies may also be different. In this manuscript, 2K1C hypertension is not as severe as Ang II-infused hypertension, which may explain the small differences in the heart and body wt. ratios between WT (0.50) vs. WT-2K1C mice (0.53).
Reviewer #2's minor comment #4: Definitions of “Control” in Figure 3, 4 and 5 are missing. Are they sham-operated mice?
Authors’ responses: They are sham controls without clipping the left renal artery.
Reviewer #2's minor comment #5: Units of Na, K, Cl excretion in Table 1 would be incorrect.
Authors’ responses: Thanks for pointing out the errors in the units of Na+, K+, Cl- excretion shown in Tables. They should be in the units of µmol/24 h.
Reviewer #2's minor comment #6: Legends of Figure 4B and 4C are oppositely described.
Authors’ responses: Thanks for pointing out the errors, which have been corrected as suggested.
Reviewer 3 Report
The proposed manuscript is very comprehensive and with a clear goal.
In this study, the authors evaluated the genetic deletion of the AT1a receptor or Na+/H+ exchanger 3 selectively in the proximal tubules of the kidney attenuates two kidneys, one‐clip Goldblatt hypertension in mice. Accordingly, the authors confirmed that the present study provides new evidence for the critical role of prox-28 imal tubule Ang II/AT1 (AT1a)/NHE3 signaling in the development of 2K1C Goldblatt hypertension.
In general, the manuscript is well written, so the reviewer found some technical errors (please see enclosed PDF file).
All research activities were performed in detail.
Statistical processing of the results and their categorization were also systematically performed.
Material and methods: Very well written.
Results: This section is excellently written with all the necessary and concise accompanying explanations.
The results correspond to the objectives of the study.
In the tables, all results are presented and easy to compare.
The figures are in a satisfactory resolution so that all the mentioned and explained details are visible.
Discussion: The discussion part is included in the results and explained to a completely satisfactory extent.
The references used are carefully selected and also up-to-date.

Author Response
Authors’ Responses to Reviewer #3: Many thanks for your expert comments and suggestions on our manuscript. We have made the changes as you suggested throughout the manuscript to further improve our manuscript.
Round 2
Reviewer 2 Report
Reviewer #2's major comment #5: Renal mRNA expression of angiotensinogen and ACE in wild type mice was not affected in the development of 2K1C hypertension. These results suggest that circulating RAS rather than renal tissue RAS plays a critical role in this model. Overall, in the pathophysiology of 2K1C hypertension, which is critical? Or both? Please clarify this point.
Authors’ responses: Thanks for the comments and your expert insights. We believe that both circulating and renal tissue RAS contribute to the pathophysiology of 2K1C hypertension, but renal tissue RAS perhaps more critical in our views. This is because the development of 2K1C hypertension in wild-type mice or humans originates from the renal artery stenosis that first activates renal tissue RAS, especially renal renin mRNA expression and active renin release into the clipped kidney and the circulation. The activation of the circulating RAS is secondary to the activation of the renal tissue RAS due to renal artery stenosis. If the renal artery stenosis is corrected, then 2K1C hypertension would not develop in the absence of renal artery stenosis.
=> Please include this discussion in the Discussion section.
Reviewer #2's minor comment #1: Methods of blood volume, Hct and urinary sodium excretion measurements are missing. Also, protocol of the Ang II infusion experiment is not described in the Methods section.
Authors’ responses: Thanks for the comments and suggestions. However, the methods for measurements of blood volume, Hct and urinary sodium excretion have been reported extensively in our previously published studies (Ref #32-35 and many others), and therefore omitted to shorten the length of the manuscript.
=> The authors can include these information as supplementary methods.
Reviewer #2's minor comment #4: Definitions of “Control” in Figure 3, 4 and 5 are missing. Are they sham-operated mice?
Authors’ responses: They are sham controls without clipping the left renal artery.
=> Please revise them accordingly so that readers can understand the results easily.
Author Response
Many thanks for your careful and kind reviews of our revised manuscript. We accept your suggestions to include the information you requested in the further revised manuscript (Red texts). Hope that they are acceptable and the manuscript is further improved.
